# MicroRNA mediated regulation in early-onset cardiac hypertrophy: Insights from the hypertrophic heart rat model

Shahzad Sadiq[1†*], Fadi J. Charchar[2], Andrew Sanigorski[3], Tamsyn Crowley[4], Hansraj R. Bookun[5‡], David N. McClure[1,6‡]

1 School of Medicine, Faculty of Health, Deakin University, Waurn Ponds, Victoria, Australia, 2 School of Health Sciences, Faculty of Science and Technology, Federation University, Ballarat, Victoria, Australia, 3 School of Health and Social Development, Faculty of Health, Deakin University, Waurn Ponds, Victoria, Australia, 4 School of Environmental and Rural Science, University of New England, Armidale, New South Wales, Australia, 5 Division of Surgery, Ballarat Base Hospital, Grampians Health, Victoria, Australia, 6 Geelong Vascular Service, Geelong, Victoria, Australia

† SS is the first author.
‡ HB and DM are joint senior authors.
* shahzad.sadiq@msn.com

## Abstract

Cardiac hypertrophy is a pathological response to increased myocardial stress and a key contributor to heart failure. The Hypertrophic Heart Rat (HHR) is a well-established model for investigating primary cardiac hypertrophy in the absence of hypertension. This study aimed to characterise the role of microRNAs and their downstream regulatory pathways in neonatal HHRs to identify mechanisms contributing to early-onset hypertrophy. Heart tissue from 2-day-old HHRs and Normal Heart Rats (NHRs) was analysed using microarray profiling to identify differentially expressed miRNAs and mRNAs. Gene Set Enrichment Analysis (GSEA) was performed to identify biological pathways associated with miRNA target genes followed by selected miRNA and genes validated by quantitative reverse transcription PCR (RT-qPCR). Neonatal HHRs demonstrated a significantly elevated cardiac weight index compared to NHRs, with upregulation of hypertrophic markers including *NPPA*, *NPPB*, and *MAPK1*. Microarray analysis revealed 107 differentially expressed miRNAs, among which miR-34a, miR-351, and miR-490* were validated and further analysed. These miRNAs were linked to key hypertrophic pathways including RAS-MAPK and PI3K-AKT along with calcium signalling. miR-34a was experimentally validated to target *HTR2A*, implicating serotonin signalling in neonatal cardiac remodelling. Additionally, elevated *SGPP1* expression suggests increased sphingolipid metabolism, while *ITGA7* was reduced and *GANC* showed a modest decrease, indicating dysregulation in mechano-signal transduction and glycogen metabolism. These findings provide insight into the early molecular drivers of cardiac hypertrophy in neonatal HHR and delineate miRNA–mRNA relationships involved in remodelling.

**Data availability statement:** All relevant data are within the manuscript and its Supporting Information files.

**Funding:** The author(s) received no specific funding for this work.

**Competing interests:** The authors have declared that no competing interests exist.

This study lays the groundwork for future investigations into the therapeutic potential of targeting miRNA pathways in the prevention and management of pathological cardiac remodelling.

## Introduction

Cardiac hypertrophy is a compensatory increase in cardiac cell mass that can lead to severe cardiovascular outcomes, including heart failure, myocardial infarction, and arrhythmias [1,2]. The Framingham Heart Study identified left ventricular hypertrophy as a critical cardiovascular risk factor, second only to age [1]. Its incidence increases with age, largely driven by hypertension and coronary heart disease, which are closely associated with left ventricular hypertrophy [3].

Cardiac hypertrophy is broadly classified as pathological or physiological. Pathological hypertrophy results from high energy demands in hypoxic environments, such as hypertension, valvular disease, or myocardial infarction [4]. It is characterised by cardiomyocyte apoptosis, necrosis, structural remodelling, and fibrosis, along with increased type I collagen deposition and reactivation of the foetal gene program (FGP), which includes Natriuretic Peptide A (*NPPA*), Natriuretic Peptide B (*NPPB*), Skeletal Alpha-Actin 1 (*ACTA1*), and Beta-Myosin Heavy Chain 7 (*MYH7*) [5,6]. This maladaptive response involves activation of pro-hypertrophic pathways, such as Rat Sarcoma-Mitogen-Activated Protein Kinase (RAS-MAPK) and Calcium dependent Calcineurin-Nuclear Factor of Activated T-cells (CN-NFAT) signalling pathway, driving transcriptional changes linked to apoptosis, metabolism and hypertrophy [4,7]. In contrast, physiological hypertrophy, induced by stressors such as pregnancy, involves reversible adaptations and preserved cardiac function [8].

Animal models have been instrumental in advancing our understanding of muscular hypertrophy. One such model, the Spontaneously Hypertensive Rat (SHR), has been widely used to study hypertension-related hypertrophy and the renin-angiotensin-aldosterone system [9]. Lately, the Hypertrophic Heart Rat (HHR); strain of *Rattus norvegicus*, emerged as a unique model for studying primary cardiac hypertrophy in the absence of hypertension [10]. Mammalian cardiomyocytes are unique for their early onset of functionality during foetal life and limited proliferative capacity postpartum [11]. As such, the HHR is characterised by increased cardiac mass, larger cardiomyocytes, and compensatory hypertrophy driven by apoptosis and decreased cardiomyocyte proliferation during neonatal development [12,13].

MicroRNAs (miRNAs) regulate hypertrophic processes by suppressing mRNA expression [14]. Dysregulated miRNA expression can cause congenital anomalies. For example, miR-1 controls cardiomyocyte cell-cycle exit by inhibiting 'Heart and Neural Crest Derivatives Expressed 2' (*HAND2*), crucial for ventricular growth and trabeculation [15]. miR-1 overexpression leads to foetal death at E13.5 in mice, while its deletion causes ventricular septal defects and prolonged mitosis underscoring its role in cardiac development [16,17]. In mature hearts, miRNA modulate hypertrophic signalling. For instance, miR-19a/b promotes CN-NFAT and Protein Kinase C (PKC)

signalling by targeting F-box protein 32 (*ATROGIN*) and Muscle RING-finger protein 1 (*MURF1*), miR-22 activates the PI3K-AKT pathway by repressing Phosphatase and tensin homolog (*PTEN*)*,* while miR-328 increases intracellular calcium and CN-NFAT signalling by suppressing Sarcoplasmic/endoplasmic reticulum calcium ATPase 2 (*SERCA2A*) [18–22].

The HHR is a genetically driven model that enables investigation of miRNA expression in primary cardiac hypertrophy without the confounding effects of hypertension. This study profiles miRNAs in neonatal HHRs and validates miRNA–mRNA interactions previously implicated in hypertrophic remodelling. The findings provide further insights into miRNA-mediated regulation and established hypertrophic pathways, highlighting potential therapeutic targets for pathological cardiac hypertrophy.

## Materials and methods

### 1. Tissue harvest

HHRs and Normal Heart Rats (NHRs) were bred and housed at Deakin University (Waurn Ponds, Australia) under conditions approved by the Deakin University Animal Ethics Committee (AECG G03-2013). Animals were maintained in standard housing conditions, with a 12-hour light/dark cycle and room temperature of 19–22℃. All animals were fed Barastoc™ Rat and Mouse Feed (Ridley, Melbourne, Australia). Male and female neonatal HHRs and NHRs aged 2 days, were culled via swift decapitation. Hearts were immediately harvested post-mortem, washed in HEPES-Krebs buffer, and non-cardiac tissues were carefully removed. Atria were dissected from the ventricles, and all samples were weighed and rapidly frozen in liquid nitrogen before storage at −80℃.

### 2. Microarray: hybridisation, acquisition, and dataset pre-processing

RNA from cardiac tissue of 2-days old HHRs and NHRs was isolated using miRNeasy (Qiagen, Australia). RNA quality was determined using the Agilent 2100 Bioanalyser (Agilent Technologies, USA) with only samples having a RNA Integrity Score (RIN) higher than "9" on 10-point scale considered for microarray analyses. Samples were hybridised to miRNA arrays (Agilent Technologies, product number G4471A, design 031189) and RNA arrays (Affymetrix™, Thermo Fisher, Australia) with microarray scanning performed by Ramaciotti Centre for Genomics (New South Wales, Australia) using standard protocols.

### 3. Functional enrichment analysis

The microarray dataset was cleaned to remove features where the processing signal was less than 2-fold variance from background and detected in at least five of the eight replicates. Overall, 2861 microRNAs and 8916 genes fulfilled the selection criteria with greater than 85% of those genes having seven or more values for each the eight replicates per treatment group. The resultant data was normalized to the 75th percentile value for each array and the p-values adjusted to allow a stringent analysis of the results.

Potential target genes for the miRNAs were determined using Ingenuity Pathway Analysis (IPA) (Qiagen, Australia) that uses predicted miRNA-mRNA relationship database of TargetScan (Whitehead Institute for Biomedical Research, Massachusetts, USA). The parameters were set as follows: reference set; 'Ingenuity Knowledge Base', relationships; 'direct' and 'indirect', species; 'rat' and prediction confidence; 'high', 'moderate' or 'experimentally observed'.

KEGG Pathways were identified using Kyoto Encyclopedia of Genes and Genomes Pathways database (Kanehisa et al., 2016). Gene set enrichment analysis (GSEA) of potential target genes was performed to examine enrichment of KEGG Pathways and Gene Ontology Biological Processes (GO:BP) using DAVID 6.8 (Database for Annotation, Visualization, and Integrated Discovery, Bioinformatics Resources, NIAID, NIH) [23,24]. Additionally, results from enrichment of GO:BP were refined by means of pathophysiological descriptors for cardiac hypertrophy. Further analysis to compare and identify KEGG Pathways and GO:BP was performed using Excel (Microsoft®). Fig 1 outlines the protocol of the target gene prediction analysis undertaken.

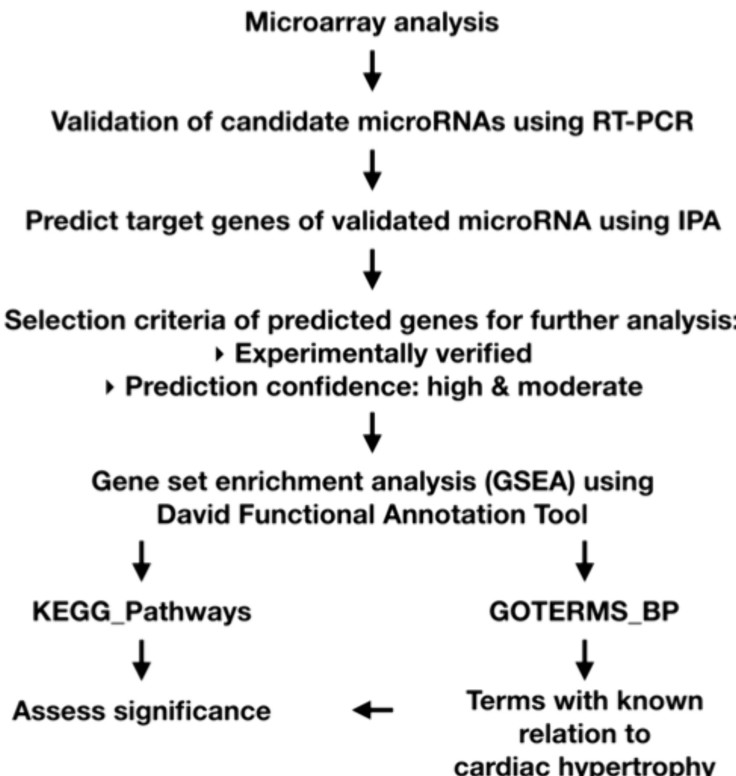

**Fig 1. Outline of steps taken in GSEA.**

MicroRNAs that were revealed to be differentially expressed between the HHR and NHR on microarray were further validated by RT-qPCR. A list of target genes of each validated microRNA was populated using IPA, with genes that were experimentally verified or had a prediction confidence score of high-moderate chosen for further analysis. GSEA was performed using DAVID to ascertain KEGG Pathways and GO:BP that may potentially be regulated by the validated miRNAs.

### 4. Quantitative reverse transcription PCR (RT-qPCR)

All steps were carried out following manufacturer's recommended protocol. Small and large RNA were isolated from ventricles HHRs and NHRs (n = 10 each), as well as transfected H9C2(2−1) cells, using mirVana™ miRNA Isolation Kit (Ambion®, Thermo Fisher, Australia). Small RNA was quantified using the Agilent Small RNA Kit and Bioanalyzer 2100 (Agilent Technologies, Australia), reverse-transcribed using the TaqMan® MicroRNA Reverse Transcription Kit (Applied Biosystems™, Thermo Fisher, Australia), and analysed with TaqMan® microRNA Assays. Large RNA was quantified using the NanoVue spectrophotometer (GE Healthcare Life Science, Australia) and reverse-transcribed with the iScript™ cDNA Synthesis Kit (Bio-Rad, Australia). The cDNA was quantified using the Quant-iT™ OliGreen® ssDNA Assay Kit (Invitrogen™, Thermo Fisher, Australia), with fluorescence measured using the Fusion-Alpha HT System (Perkin Elmer®, Australia) under the following conditions: Fluorescein 485 nm, Emission Filter 535 nm, and Light Source Intensity Level = 1.

RT-qPCR was conducted using TaqMan® Universal PCR Master Mix, No AmpErase® UNG (Applied Biosystems™, Thermo Fisher, Australia), according to the manufacturer's instructions. MicroRNA and gene expression assays were performed with 1 ng and 100 ng of cDNA per reaction, respectively. Amplification was measured using the CFX96™ Real-Time System (Bio-Rad, Australia).

### 5. Cell line transfections

H9C2(2−1) cells obtained directly from Sigma-Aldrich® (Merck, Australia) were cultured in Dulbecco's Modified Eagle Medium (DMEM) with high glucose (Sigma-Aldrich®, Merck, Australia), supplemented with 10% Foetal Bovine Serum and Antibiotic-Antimycotic Solution (Sigma-Aldrich®, Merck, Australia), in 5% $CO_2$ at 37°C. Cells were sub-cultured at 70% confluence, and transfections were performed using Lipofectamine™ RNAiMAX Transfection Reagent (Applied Biosystems™, Thermo Fisher, Australia). MiRNA mimics and inhibitors (mirVana™, Ambion®, Thermo Fisher, Australia) were diluted in Opti-MEM® Medium (Gibco™, Thermo Fisher, Australia) and mixed with lipofectamine in a 1:1 ratio. The miRNA-lipid complex was added to cultures at 70% confluence and incubated for 48 hours before RNA extraction. Protocols were followed as per the manufacturer's guidelines.

### 6. Statistical analysis

The Cardiac Weight Index (CWI) was calculated as the ratio of heart weight (mg) or ventricle weight (mg) to total body weight (g). For RT-qPCR analysis, average cycle threshold (Ct) values ± 2 standard deviations were compared between groups. Gene and microRNA expression fold changes were determined using the $2^{\wedge}(-\Delta Ct)$ method. Data are presented as mean ± standard error of the mean (SEM). Statistical analyses were conducted using GraphPad Prism 7.0, with comparisons made using Student's $t$-test, and $p < 0.05$ considered statistically significant.

## Results

### 1. Cardiac hypertrophy in the HHR

At 2 days of age, the HHR cohort demonstrated a significantly elevated heart-body weight CWI compared to the NHR cohort; 7.29 ± 0.07 mg/g vs. 6.26 ± 0.34 mg/g, ($p = 0.008$); Fig 2A. Similarly, the ventricle-body weight CWI was significantly elevated in HHRs compared to NHRs; 5.73 ± 0.14 mg/g vs 4.61 ± 0.26 mg/g, ($p = 0.002$); Fig 2B. However, no significant difference was observed in the body weights of HHRs and NHRs (Fig 2C).

Expression of established biomarkers of cardiac hypertrophy were quantified using RT-qPCR. *NPPA* ($p = 0.0218$), *NPPB* ($p = 0.0004$) and Mitogen-Activated Protein Kinase 1 (*MAPK1*) ($p = 0.0005$) levels were significantly elevated in HHRs compared to controls. In contrast, no statistical differences were observed in the expression of *MYH7* or *ACTA1*. Fig 3 summarises the fold changes in the expression of these cardiac biomarkers.

### 2. miRNA expression

Microarray analysis identified 107 miRNAs as significantly upregulated or downregulated, based on an unadjusted p-value of $p < 0.05$ (Table 1), detailed results provided in supplementary S1A Table in S1 File. Of these, the eight miRNAs with the lowest p-values were selected for further investigation. Seven of the eight selected miRNAs were upregulated in 2-day-old HHRs, while miR-466b was uniquely downregulated, with a fold change of −54.65. Among the upregulated miRNAs, miR-34a exhibited the highest positive fold change of 2.45 ($p = 3.07e$-07). Notably, three of the identified miRNAs; miR-378*, miR-490*, and miR-675*, are classified as non-dominant, as indicated by the asterisk in their nomenclature, reflecting limited prior reports of their expression.

RT-qPCR validation of the eight selected miRNAs confirmed significant regulation of miR-34a, miR-378*, miR-490* and miR-351 (Table 2). Fold changes for this set of miRNAs mostly aligned with microarray findings, except for miR-490, which was downregulated. Despite substantial downregulation of miR-466b in the microarray analysis, RT-qPCR showed no significant change. Discrepancies between the two widely used transcriptome analysis methods may be attributed to the lower sensitivity of microarrays relative to RT-qPCR, particularly for targets with low baseline expression levels.

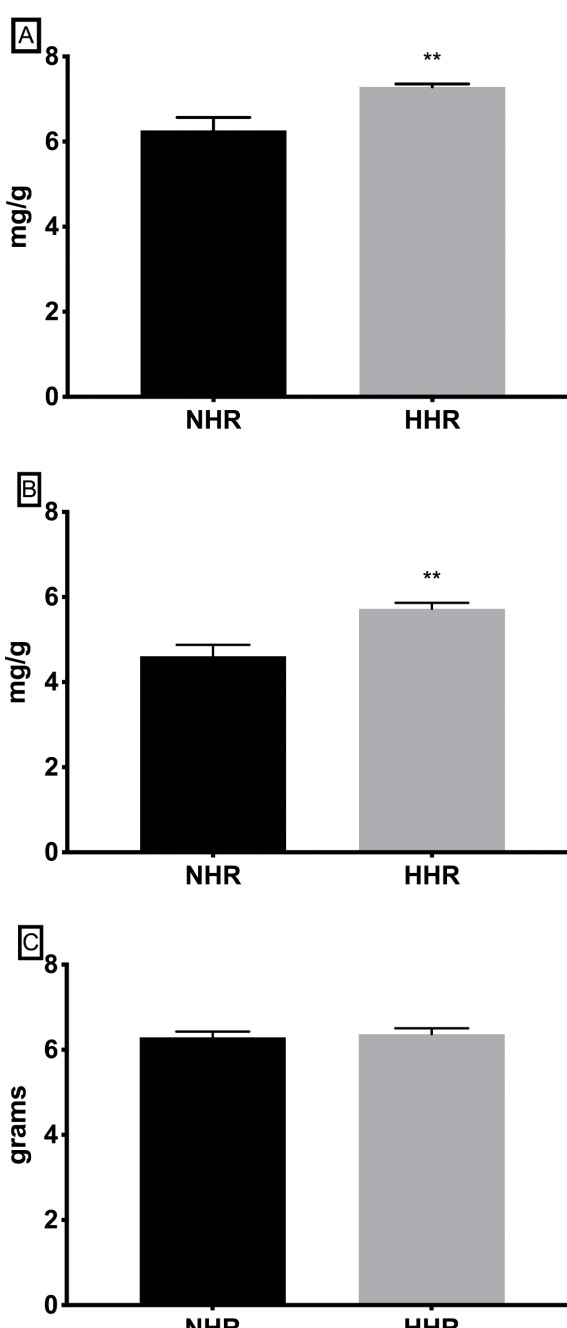

**Fig 2. Results of cardiac phenotypical measurements.** Heart, ventricle and body weights were measured and cardiac weight (mg) to body (g) indices were calculated. At 2-days of age, (A) heart-body (mg/g) ratio and (B) ventricle-body (mg/g) ratio were significantly higher in the HHR, which is suggestive of an increased heart and ventricle mass. (C) The body weights of HHR and NHR were similar at 2-days of age. Two asterisks (**) denote significance at $p < 0.01$.

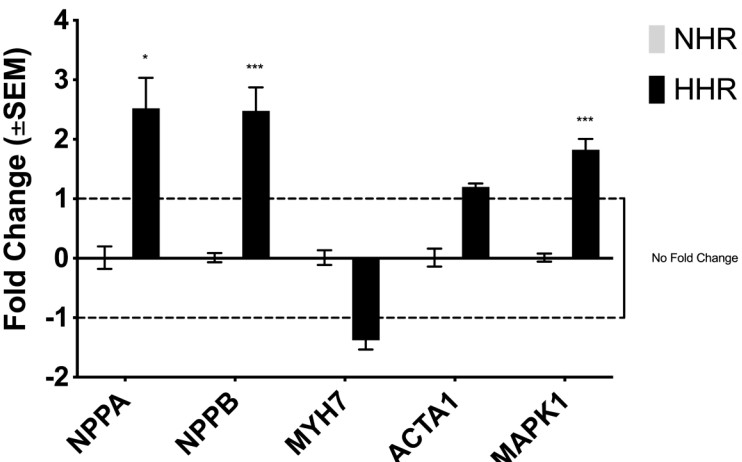

**Fig 3. Expression of genetic biomarkers of cardiac hypertrophy in the HHRs.** Expression of genetic biomarkers associated with cardiac hypertrophy in HHR hearts are presented as fold changes relative to controls. *NPPA* and *NPPB* exhibited fold changes greater than 2.0, while *MAPK1* showed a fold change exceeding 1.5. Elevated *NPPA* and *NPPB* levels indicate myocardial wall stress and serve a cardio-protective function. *MAPK1*, a key downstream regulator, is involved in the pro-hypertrophic MAPK signalling pathway. In contrast, the fold changes in *MYH7* and *ACTA1* expression were not statistically significant. One asterisk (*) and three asterisks (***) denote significance at $p < 0.05$ and $p < 0.001$, respectively.

### 3. Gene set enrichment analysis (GSEA)

Initial target prediction analysis using IPA identified several thousand potential target genes for each miRNA. miR-378* was excluded from further analysis due to a fold change of less than 1.5 on RT-qPCR. In summary, miR-34a had 1,476 potential target genes, miR-351 had 1,531, and miR-490* had 384, (detailed results are provided in S2 Tables in S1 File). Notably, the analysis revealed no experimentally verified target genes for miR-490* in *Rattus norvegicus*. However, one previously verified target gene, *FBJ murine osteosarcoma viral oncogene homolog* (*c-FOS*), is implicated in cellular proliferation in human bladder cancer [25].

### 4. Enriched pathways

The three miRNAs demonstrate overlapping regulatory pathways, including neuronal axon guidance, oxytocin signalling, and RAS signalling. Shared pathways between miR-34a and miR-351 include those related to cancers such as prostate cancer, melanoma, and chronic myeloid leukaemia, as well as pathways involved in cellular metabolism, endocytosis, and key signalling cascades, such as MAPK, phosphoinositide-3-kinase (PI3K-AKT), wingless-type MMTV integration site family (WNT), and adrenergic signalling. In contrast, pathways shared between miR-34a and miR-490* include thyroid hormone synthesis, oestrogen signalling, and circadian rhythm regulation. Supplementary S1 Fig in S1 File presents a numerical comparison of KEGG pathways enriched by the target genes of miR-34a, miR-351, and miR-490*.

When assessed individually, miR-34a predominantly regulates pathways linked to cancer, signalling mechanisms, and cardiomyopathy. miR-351 modulates pathways associated with cancer, metabolism, and signalling processes, as well as viral infection pathways, including measles and hepatitis B and C. miR-490* is primarily linked to signalling mechanisms, infections, purine metabolism, and endocrine function.

Grouping KEGG pathways by physiological and pathological descriptors provides further insight into the potential roles of these miRNAs, as illustrated in Fig 4, with additional details available in supplementary S2 Tables in S1 File. Table 3 lists pathways associated with cardiac hypertrophy that are potentially regulated by these miRNAs.

**Table 1. Summary of results from microarray analysis.**

| Probeset ID | Un-adjusted p-value | Fold Change(HHR/NHR) |
|---|---|---|
| rno-miR-34a | 3.07e-07 | 2.45 |
| rno-miR-378* | 8.87e-06 | 1.63 |
| rno-miR-378 | 9.52e-06 | 1.68 |
| rno-miR-218 | 7.17e-05 | 1.64 |
| rno-miR-466b | 1.19e-04 | −54.65 |
| rno-miR-490* | 1.76e-04 | 1.47 |
| rno-miR-675* | 1.66e-04 | 1.47 |
| rno-miR-351 | 2.08e-04 | 1.48 |
| rno-miR-674-3p | 2.40e-04 | 1.48 |
| rno-miR-3085 | 3.00e-04 | −1.13 |
| rno-miR-214 | 3.45e-04 | 1.40 |
| rno-miR-410 | 5.21e-04 | 1.66 |
| rno-miR-503 | 6.24e-04 | 1.45 |
| rno-miR-199a-5p | 9.37e-04 | 1.45 |
| rno-miR-490 | 1.09e-03 | 1.39 |
| rno-miR-431 | 1.02e-03 | 1.57 |
| rno-miR-199a-3p | 2.18e-03 | 1.37 |
| rno-miR-542-5p | 2.65e-03 | 1.36 |
| rno-miR-329 | 3.71e-03 | 1.51 |
| rno-miR-300-3p | 3.54e-03 | 1.52 |
| rno-miR-423* | 3.97e-03 | 14.32 |
| rno-miR-483 | 4.17e-03 | 1.16 |
| rno-miR-542-3p | 4.35e-03 | 1.32 |
| rno-miR-322* | 4.59e-03 | 1.36 |
| rno-miR-500 | 5.56e-03 | −1.30 |
| rno-miR-532-3p | 5.44e-03 | −1.29 |
| rno-miR-223 | 7.10e-03 | −1.10 |
| rno-miR-484 | 6.98e-03 | 1.24 |
| rno-miR-1949 | 6.94e-03 | 1.28 |
| rno-miR-18a | 7.03e-03 | 1.33 |
| rno-miR-106b* | 7.54e-03 | 1.29 |
| rno-miR-450a | 7.64e-03 | 1.33 |
| rno-let-7b | 9.31e-03 | 1.23 |
| rno-let-7c | 9.05e-03 | 1.23 |
| rno-miR-335 | 8.73e-03 | 1.32 |
| rno-miR-487b | 8.56e-03 | 1.48 |
| rno-miR-341 | 9.14e-03 | 12.07 |
| rno-miR-3584-5p | 1.36e-02 | −49.79 |
| rno-miR-362 | 1.32e-02 | −1.36 |
| rno-miR-92a | 1.22e-02 | 1.21 |
| rno-miR-99b | 1.06e-02 | 1.22 |
| rno-let-7e | 1.30e-02 | 1.22 |
| rno-miR-128 | 1.14e-02 | 1.25 |
| rno-miR-99a | 1.26e-02 | 1.26 |
| rno-miR-20a | 1.33e-02 | 1.28 |

*(Continued)*

**Table 1.** (Continued)

| Probeset ID | Un-adjusted p-value | Fold Change(HHR/NHR) |
|---|---|---|
| rno-miR-19b | 1.00e-02 | 1.29 |
| rno-miR-19a | 1.08e-02 | 1.29 |
| rno-miR-434 | 1.32e-02 | 1.31 |
| rno-miR-127 | 1.30e-02 | 1.32 |
| rno-miR-101b | 1.40e-02 | 1.32 |
| rno-miR-379 | 1.09e-02 | 1.38 |
| rno-miR-136 | 1.30e-02 | 1.59 |
| rno-miR-31 | 1.39e-02 | 9.60 |
| rno-miR-598-3p | 1.04e-02 | 11.58 |
| rno-miR-342-3p | 1.59e-02 | 1.21 |
| rno-miR-185 | 1.63e-02 | 1.26 |
| rno-miR-17-5p | 1.60e-02 | 1.29 |
| rno-miR-411* | 1.62e-02 | 1.34 |
| rno-miR-338 | 1.53e-02 | 7.75 |
| rno-miR-466b-1* | 1.66e-02 | −1.22 |
| rno-miR-92b | 1.77e-02 | −1.32 |
| rno-miR-328a* | 1.95e-02 | −9.66 |
| rno-miR-20b-5p | 1.92e-02 | 1.28 |
| rno-miR-376a | 1.91e-02 | 1.63 |
| rno-miR-211* | 1.98e-02 | −9.87 |
| rno-miR-130a | 2.12e-02 | 1.23 |
| rno-miR-652 | 2.36e-02 | 1.20 |
| rno-miR-133a* | 2.33e-02 | 1.22 |
| rno-miR-361 | 2.46e-02 | 1.22 |
| rno-miR-30e* | 2.50e-02 | 1.23 |
| rno-let-7f | 2.68e-02 | 1.21 |
| rno-miR-133a | 2.83e-02 | 1.21 |
| rno-miR-26b | 2.81e-02 | 1.22 |
| rno-miR-455* | 2.92e-02 | 1.13 |
| rno-miR-30a* | 2.97e-02 | 1.22 |
| rno-miR-101a | 2.91e-02 | 1.24 |
| rno-miR-347 | 2.98e-02 | 1.35 |
| rno-miR-106b | 3.13e-02 | 1.22 |
| rno-miR-144 | 3.10e-02 | 1.32 |
| rno-miR-140 | 3.22e-02 | 1.20 |
| rno-miR-324-5p | 3.21e-02 | 1.22 |
| rno-miR-32* | 3.38e-02 | −15.67 |
| rno-miR-150 | 3.46e-02 | −1.23 |
| rno-miR-3573-3p | 3.38e-02 | −1.09 |
| rno-miR-151 | 3.51e-02 | 1.19 |
| rno-miR-331 | 3.49e-02 | 1.20 |
| rno-miR-322 | 3.54e-02 | 1.21 |
| rno-miR-221 | 3.48e-02 | 10.69 |
| rno-miR-125b-5p | 3.59e-02 | 1.20 |
| rno-miR-28 | 3.65e-02 | 1.21 |

*(Continued)*

**Table 1.** (Continued)

| Probeset ID | Un-adjusted p-value | Fold Change(HHR/NHR) |
|---|---|---|
| rno-miR-25 | 3.73e-02 | 1.21 |
| rno-miR-1 | 3.79e-02 | 1.18 |
| rno-miR-98 | 3.86e-02 | 1.21 |
| rno-miR-15b | 3.97e-02 | 1.18 |
| rno-miR-93 | 3.99e-02 | 1.22 |
| rno-miR-455 | 3.99e-02 | 1.25 |
| rno-miR-324-3p | 4.28e-02 | 1.28 |
| rno-miR-451 | 4.27e-02 | 1.29 |
| rno-miR-107 | 4.34e-02 | 1.20 |
| rno-miR-21* | 4.43e-02 | 5.70 |
| rno-miR-382 | 4.40e-02 | 7.99 |
| rno-miR-3588 | 4.54e-02 | −12.24 |
| rno-miR-130b | 4.77e-02 | 1.19 |
| rno-miR-148b-3p | 4.80e-02 | 1.20 |
| rno-miR-30b-5p | 4.64e-02 | 1.21 |
| rno-miR-152 | 4.78e-02 | 1.23 |
| rno-miR-29b | 4.68e-02 | 1.26 |

**Table 2. Summary of results from RT-PCR validation.**

| Probe Set ID | RT-PCR | |
|---|---|---|
| | p-value | Fold Change (HHR/NHR) |
| rno-miR-34a | 0.0013** | 1.77 |
| rno-miR-378* | 0.0239* | 1.31 |
| rno-miR-378 | 0.0780 | −2.23 |
| rno-miR-218 | 0.1944 | −1.39 |
| rno-miR-466b | 0.2373 | −1.57 |
| rno-miR-490* | 0.0327* | −2.35 |
| rno-miR-675* | 0.1967 | −1.78 |
| rno-miR-351 | 0.0095** | 1.96 |

Significantly expressed miRNAs from the microarray were ranked by ascending p-values with corresponding fold changes. The top eight miRNAs were further validated by RT-qPCR, confirming significant differential expression of miR-34a, miR-378*, miR-490*, and miR-351 in HHRs. Except for miR-490*, fold change trends of the significant miRNAs were consistent between the two validation methods. One asterisk (*) and two asterisks (**) denote significance at $p < 0.05$ and $p < 0.01$, respectively.

The graph summarises KEGG Pathways modulated by miR-34a, miR-351 and miR-490* that have been categorised using physiological and pathological descriptors of cardiac hypertrophy.

## 5. Biological processes

We identified several hundred GO:BP terms for each miRNA, with detailed results provided in supplementary S2 Tables in S1 File. These terms were filtered using keywords relevant to the pathogenesis of cardiac hypertrophy and a comparison of the number of GO:BP identified for each microRNA is shown in supplementary S2 Fig in S1 File and detailed in

**Fig 4. KEGG Pathway of cardiac hypertrophy regulated by miR-34a, miR-351 and miR-490*.**

**Table 3. KEGG pathways associated with cardiac hypertrophy that are potentially regulated by miRNAs under investigation.**

| KEGG Pathway | Entry | miR-34a | miR-351 | miR-490* |
|---|---|---|---|---|
| RAS signalling pathway | rno04014 | X | X | X |
| Oestrogen signalling pathway | rno04915 | X | | X |
| Adrenergic signalling in cardiomyocytes | rno04261 | X | X | |
| MAPK signalling pathway | rno04010 | X | X | |
| PI3K-AKT signalling pathway | rno04151 | X | X | |
| Calcium signalling pathway | rno04020 | | X | |
| Thyroid hormone signalling pathway | rno04919 | X | | |
| Hypertrophic cardiomyopathy (HCM) | rno05410 | X | | |
| Dilated cardiomyopathy (DCM) | rno05414 | X | | |

supplementary S2 Tables in S1 File. Fig 5 summarises the biological processes relevant to cardiac hypertrophy, which is the focus of this study.

The graph summarises various biological processes being modulated by miR-34a, miR-351 and miR-490* that have been categorised using descriptors associated with development of cardiac hypertrophy.

Biological processes collectively regulated by miR-34a, miR-351, and miR-490* include cardiac morphogenesis, cellular proliferation, apoptosis, and stress-induced MAPK signalling. Individually, miR-34a is associated with diverse functions, including cardiogenesis, calcium handling, apoptosis, muscle contraction, MAPK signalling, and angiogenesis, suggesting a central role in maintaining cardiovascular homeostasis. miR-351 contributes to apoptosis, cardiogenesis, cell size regulation, and MAPK signalling, while miR-490* is predominantly involved in cardiac conduction and muscle contraction. Together, these findings underscore the complex regulatory roles of these microRNAs in cardiomyocyte function and their potential contribution to the development of cardiac hypertrophy.

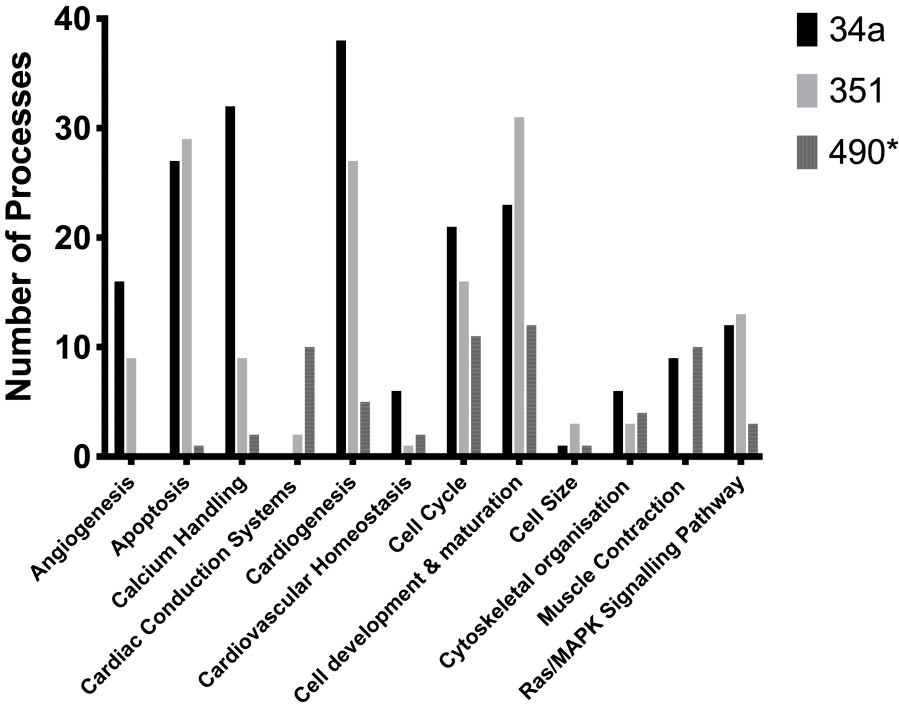

**Fig 5. Summary of GO:BP regulated by miR-34a, miR-351 and miR-490\*.**

## 6. Gene expression in HHR

Microarray analysis identified 373 genes as differentially expressed (p-value <0.05) in the heart tissue of 2-day-old HHRs (additional details provided in supplementary S1B Table in S1 File). A comparative analysis of microarray-validated genes and predicted microRNA targets revealed that miR-34a regulates 21 genes, overlapping with 2 miR-490\* targets and 1 miR-351 target. miR-351 targets 24 genes, while miR-490\* targets 12, with 1 gene shared between them. Detailed results are tabulated in supplementary S2 Table in S1 File and illustrated in supplementary S3 Fig in S1 File.

Each predicted gene target was assessed for its potential role in cardiac hypertrophy, resulting in the selection of 10 genes for further validation using RT-qPCR (Table 4). The expression levels of 5-Hydroxytryptamine Receptor 2A *(HTR2A)* (p = 0.0006) and Sphingosine-1-Phosphate Phosphatase 1 *(SGPP1)* (p = 0.0005), were significantly upregulated with fold changes exceeding 2.0. In contrast, Integrin Subunit Alpha 7 *(ITGA7)* was downregulated (fold change −1.80; p < 0.0001). For Glucosidase Alpha, Neutral C *(GANC)*, RT-qPCR indicated a modest reduction (FC −1.47; p = 0.0492), whereas the microarray estimated an increase (FC + 1.45); see Discussion for cross-platform comparison. Given the small effect size and cross-platform discordance, expression level of *GANC* was considered a nominal finding and interpreted cautiously.

## 7. Functional annotation of validated genes

Further analysis of the validated genes identified several KEGG pathways that these genes potentially modulate in the hearts of 2-day-old HHRs (Table 5).

*HTR2A* is linked to five pathways involving calcium signalling, neurological signalling, and TRP (Transient Receptor Potential) channel regulation. *SGPP1* is associated with sphingolipid metabolism and signalling, while *GANC* is linked to

**Table 4. Results of Microarray Analysis and PCR Validation of Candidate Genes in 2-Day-Old HHR Hearts.**

| Gene Symbol | microRNA | | | Microarray | | RT-PCR | |
|---|---|---|---|---|---|---|---|
| | 34a | 351 | 490* | p-value | Fold Change | p-value | Fold Change |
| HTR2A | X | | | 3.59e-06 | −3.18 | 0.0006* | 2.38 |
| PKIA | X | | | 1.17e-02 | −1.58 | 0.4096 | 1.21 |
| SGPP1 | X | | X | 4.26e-05 | −1.58 | 0.0005* | 2.16 |
| RGS6 | X | X | | 8.23e-03 | −1.55 | 0.2534 | 1.27 |
| CACNA2D2 | X | | | 3.55e-05 | −1.52 | 0.0745 | 1.37 |
| CDH13 | X | | | 3.99e-03 | −1.51 | 0.6282 | 1.09 |
| CXADR | | | X | 1.82e-03 | −1.40 | 0.3059 | 1.16 |
| AXL | X | | | 4.71e-03 | −1.32 | 0.1202 | 1.26 |
| GANC | | X | X | 4.10e-03 | 1.45 | 0.0492* | −1.47 |
| ITGA7 | | X | | 3.73e-06 | 1.88 | <0.0001* | −1.80 |

The table summarises the results of microarray and RT-qPCR analyses for 10 differentially expressed candidate genes in 2-day-old HHR hearts; *HTR2A*, protein kinase inhibitor alpha (*PKIA*), *SGPP1*, regulator of G-protein signalling 6 (*RGS6*), calcium voltage-gated channel auxiliary subunit alpha2delta 2 (*CACNA2D2*), cadherin 13 (*CDH13*), coxsackievirus and adenovirus receptor (*CXADR*), AXL receptor tyrosine kinase (*AXL*), *GANC* and *ITGA7*. Bioinformatics analysis suggests potential regulation of these genes by specific miRNAs, denoted by a 'X'.

**Table 5. Biological Pathways Associated with Genes Validated by RT-PCR.**

| Genes | KEGG Pathway ID | KEGG Pathway |
|---|---|---|
| HTR2A | rno04020 | Calcium signalling pathway |
| | rno04080 | Neuroactive ligand-receptor interaction |
| | rno04540 | Gap junction |
| | rno04726 | Serotonergic synapse |
| | rno04750 | Inflammatory mediator regulation of TRP channels |
| SGPP1 | rno00600 | Sphingolipid metabolism |
| | rno04071 | Sphingolipid signalling pathway |
| GANC | rno00052 | Galactose metabolism |
| | rno00500 | Starch and sucrose metabolism |
| | rno01100 | Metabolic pathways |
| ITGA7 | rno04151 | PI3K-Akt signalling pathway |
| | rno04510 | Focal adhesion |
| | rno04512 | ECM-receptor interaction |
| | rno04810 | Regulation of actin cytoskeleton |
| | rno05165 | Human papillomavirus infection |
| | rno05410 | Hypertrophic cardiomyopathy (HCM) |
| | rno05412 | Arrhythmogenic right ventricular cardiomyopathy (ARVC) |
| | rno05414 | Dilated cardiomyopathy |

Table listing biological pathways in which genes validated by RT-qPCR are significantly involved, highlighting their functional roles in key cellular and molecular processes.

three pathways related to carbohydrate metabolism. *ITGA7* is involved in eight pathways, including hypertrophic cardiomyopathy (HCM), arrhythmogenic right ventricular cardiomyopathy (ARVC), dilated cardiomyopathy (DCM), PI3K-AKT signalling, and actin cytoskeleton regulation.

## 8. miRNA target validation

H9c2(2–1) cells were transfected with mimics and inhibitors of miR-34a, miR-351, and miR-490*, with successful transfection confirmed by RT-qPCR (supplementary S4 Fig in S1 File). RT-qPCR results showed that *HTR2A* expression was significantly downregulated in cells transfected with the miR-34a mimic, with a fold change of −3.93 (±0.32) relative to the control ($p = 0.0005$), see Fig 6. Additionally, *HTR2A* expression was significantly lower ($p = 0.03$) in cells overexpressing miR-34a compared to those with the microRNA inhibitor. In contrast, cells treated with the miR-34a inhibitor showed an increased fold change of 1.49 (±0.34) relative to the control, though this was not statistically significant ($p = 0.65$). These results suggest that *HTR2A* is regulated by miR-34a. In contrast, no significant changes in *SGPP1* expression were observed across the treatment groups.

miR-351 was predicted to regulate *GANC* and *ITGA7*, with RT-qPCR demonstrating a two-fold increase in the expression of both genes in cells treated with the mimic; however, these changes were not statistically significant. Similarly, *GANC* was identified as a potential target of miR-490, and RT-qPCR analysis showed its downregulation in cells transfected with the miR-490 mimic, though this result was also not statistically significant. Notably, there was considerable variability in the expression measurements for miR-351 and miR-490* target genes, which may have contributed to the lack of statistical significance despite observed fold changes. These findings were inconclusive and warrant further investigation.

## Discussion

This study characterises differential expression of miR-34a, miR-351, and miR-490* in neonatal HHRs, defining a distinct miRNA signature and associated pathways. The findings generalise established hypertrophic mechanisms to a normotensive, genetically predisposed model at an early developmental stage.

miR-34a is consistently implicated in maladaptive ventricular remodelling, hypertrophy and fibrosis, acting through anti-apoptotic and stress-response pathways that include direct repression of Sirtuin 1 (*SIRT1*), Aldehyde Dehydrogenase 2 (*ALDH2*), B-cell CLL/Lymphoma 2 (*BCL2*), and Phosphatase 1 Nuclear Targeting Subunit (*PNUTS*), besides with links to telomere length regulation and ageing [13,26–28]. Therapeutic inhibition of the miR-34 family can attenuate remodelling, though efficacy is variable across models, with only modest benefit reported in established type 1 diabetic

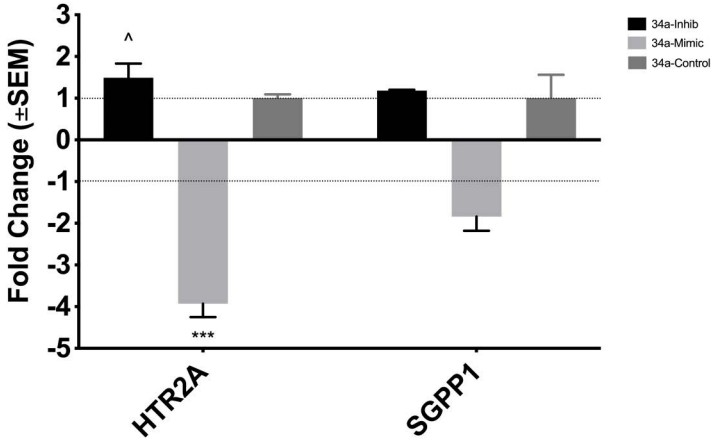

**Fig 6. Expression of potential target genes of miR-34a in H9C2(2-1) following transfection with mimic and inhibitor.** In cells that were transfected with a mimic, expression of *HTR2A* is downregulated by a FC of −3.93, compared to the control. Similar results can be seen when expression of *HTR2A* is compared between cells with mimic and inhibitor of miR-34a ($p = 0.03$). The expression of *SGPP1* is not statistically significant across the treatment groups and control. *** refers to $p < 0.001$, ^ refers to $p < 0.05$.

cardiomyopathy [26,29]. Moreover, following myocardial infarction, miR-34a expression increases in the injured myocardium, where as its inhibition improves ventricular function, reduces fibrosis and cell death, and enhances myocardial repair [30].

miR-351 is linked to hypertrophic responses in experimental models. In mice, the long non-coding RNA *LncSync* coordinates with miR-351 to regulate cardiomyocyte homeostasis and hypertrophic growth [31]. In phenylephrine-stimulated cardiomyocytes, miR-351 targets E2F Transcription Factor 3 *(E2F3),* increasing proliferation and apoptosis resistance and thereby facilitating hypertrophic growth [32,33]. miR-490 inhibits cellular proliferation by targeting *c-FOS*, which is upregulated during phenylephrine-induced hypertrophy via the RAS pathway [25,34]. miR-490* constrains proliferative capacity in glioblastoma by repressing telomere-maintenance genes *Telomeric Repeat-binding Factor 2 (TERF2)*, *Tankyrase 2 (TNKS2)* and *Serine/Threonine-protein kinase, (SMG1)*, inducing telomere dysfunction and DNA-damage signalling [35]. GSEA combined with RT-qPCR identified four key genes, including *HTR2A*, which is demonstrated in this study to be modulated by miR-34a. These findings highlight the roles of miRNAs in apoptosis, cell proliferation, and hypertrophic remodelling, contributing to a deeper understanding of early-onset hypertrophy mechanisms in HHRs.

## 1. Cardiac hypertrophy in neonatal HHRs

The HHR represents a unique and well-established model for investigating the intrinsic drivers of primary pathological cardiac hypertrophy associated with hypertension, despite the absence of elevated blood pressure [10,36]. Previous studies have reported the overexpression of genes such as DnaJ Heat Shock Protein Family (Hsp40) Member A3 *(DNAJA3)*, G Protein Subunit Gamma 5 *(GNG5)*, RAB12, Member RAS Oncogene Family *(RAB12)*, *MAPK1*, and *NPPB*; all linked to the pro-hypertrophic RAS-MAPK signalling pathway in adult HHRs [36]. A proposed mechanism suggests that increased foetal apoptosis and reduced cardiomyocyte proliferation result in a smaller cardiac cell population at birth, driving compensatory hypertrophy later in life [12]. Additionally, dysregulated PI3K-AKT activity and reduced Mitogen-Activated Protein Kinase 14 *(MAPK14)* signalling during foetal cardiac development highlight the complexity of these regulatory pathways [12].

Neonatal HHRs showed an increased CWI compared to NHRs, suggesting a prenatal onset of compensatory remodelling, differing from prior reports from Porrello et al., (2009) and Marques et al., (2016). Gross examination revealed no structural defects, and normal growth patterns were observed. Gene analysis revealed significant upregulation of *NPPA* and *NPPB*, indicating myocardial stress and remodelling, while *MYH7* and *ACTA1* showed no changes, likely due to rapid transcription during early development. Additionally, *MAPK1*, a RAS-MAPK pathway mediator, was upregulated, driving pro-hypertrophic gene expression [37]. The findings suggest myocardial hypertrophy in neonatal HHRs with increased CWI and upregulated markers. However, potential genetic drift and inbreeding instability may contribute to observed differences, as such changes often manifest over generations, a recognised limitation of animal models [38,39].

## 2. Potential pathways implicated in cardiac hypertrophy

GSEA identified several cellular pathways and biological processes potentially regulated by miR-34a, miR-351, and miR-490*, communally or individually. These pathways, many of which are well-established in the pathogenesis of cardiac hypertrophy, are summarised in Table 3.

The RAS signalling pathway is well-established as pro-hypertrophic, driving pathological myocardial remodelling [40]. Studies on MAPK signalling cascades, including extracellular signal-regulated kinases (*ERKs*), c-Jun N-terminal kinase (*JNK*), and *MAPK14*, indicate that their overexpression collectively contributes to pathological remodelling [41]. In contrast, the PI3K-AKT pathway exerts a cardio-protective role by promoting cardiac cell growth, mitigating apoptosis, and reducing fibrosis [42]. Adrenergic signalling pathways, mediated through α- and β-adrenergic receptors, are also integral to cardiac function. Notably, activation of α1B-AR and knockout of β1/β2-AR receptors have been shown to induce significant cardiac hypertrophy [43,44]

The calcium signalling pathway plays a central role in hypertrophy, as increased intracellular calcium levels—primarily driven by heightened sarco/endoplasmic reticulum Ca²⁺-ATPase (SERCA) pump activity activate the pro-hypertrophic calcineurin-nuclear factor of activated T-cells (NFAT) pathway [7,45].

The renin-angiotensin-aldosterone system (RAAS) has emerged as a significant pathway in cardiovascular physiology, although its multifactorial nature complicates efforts to isolate its specific contributions to cardiac hypertrophy [46]. miR-490 is predicted to regulate pathways involved in renin secretion, while miR-34a is associated with aldosterone synthesis, secretion, and sodium reabsorption.

Oestrogen signalling may confer cardio-protection by activating the PI3K-AKT pathway, reducing p53-mediated apoptosis, and inhibiting calcineurin and p38-MAPK-dependent cardiac remodelling [47]. miR-34a is also speculated to influence thyroid hormone signalling, which regulates cardiac function, physiological myocardial hypertrophy, and foetal gene expression [48]. The p53 signalling pathway, another potentially key target of miR-34a, plays a crucial role in supporting DNA repair, inhibiting cell proliferation, and promoting apoptosis [49]. Notably, neonatal HHRs exhibit reduced *p53* expression, possibly as a compensatory response to decreased cardiomyocyte pools at birth [12]. Additionally, miR-34a may influence cardiac chamber development and foetal maturation, where its dysregulation could contribute to congenital heart defects, such as Tetralogy of Fallot [50,51].

A potential link between miR-34a and long-term depression pathways has also been proposed, as autonomic dysfunction and genetic predisposition have previously been identified as mechanisms connecting depression and heart disease [52]. miR-34a expression is significantly downregulated in the brains of individuals who committed suicide, warranting further investigation [53].

GSEA identified novel pathways, such as axon guidance, oxytocin signalling, sphingolipid signalling, stem cell pluripotency, and circadian rhythm regulation, expanding the understanding of cardiac hypertrophy and offering directions for future research.

## 3. Potential role of validated genes in cardiac hypertrophy

**HTR2A.** The role of serotonin as a neurotransmitter in the central nervous system is well-established. However, its pathophysiological functions in the heart, particularly myocardial remodelling and cardiogenesis, are less clearly understood [54]. Serotonin interacts with a range of 5-hydroxytryptamine (5-HT) receptors, which are targeted by drugs used to treat psychiatric disorders such as depression and psychosis [55]. The specific role of these receptors in cardiac hypertrophy remain incompletely defined.

Accumulating evidence indicates *HTR2A* engages pro-hypertrophic signalling via the PI3K–PDK1–AKT–mTOR pathway in an isoproterenol (ISO)–induced mouse model of cardiac hypertrophy [56]. Moreover, it has been demonstrated that *HTR2A* is upregulated in cardiac muscle and cardiomyocytes isolated from the left ventricle of rats subjected to transverse aortic banding, a model of pressure-overload-induced cardiac hypertrophy [57]. Serotonin exposure resulted in enhanced inotropic responses in papillary muscles, proportional to the degree of hypertrophy. This suggests that activation of the 5-HT2A receptor may improve cardiac function while reducing energy demands during hypertrophy [57]. Similar findings were observed in a rat model of acute congestive heart failure [58]

Additional evidence suggests that *HTR2A* plays a role in cardiac remodelling associated with hypertension. One study reported a 25-fold increase in *HTR2A* expression in SHR [59]. Moreover, polymorphisms in *HTR2A*, such as *T102C*, have been associated with primary hypertension [60]. A more recent study also reported associations between several *HTR2A* single nucleotide polymorphisms (SNPs), including rs6313/T102C, and hypertension risk, though the direction and magnitude of effect varied between cohorts [61]. Another study linked the *1438A/G* genotype of *HTR2A* to variations in depression severity among men with coronary heart disease [62].

Extending prior reports implicating HTR2A in hypertrophy, this study demonstrates upregulation of *HTR2A* in 2-day-old HHR hearts and identifies the gene as a regulatory target of miR-34a. Transfection of H9C2(2−1) cells with a miR-34a

mimic significantly downregulated *HTR2A* expression, suggesting a direct regulatory relationship. However, despite miR-34a upregulation in neonatal HHRs, *HTR2A* expression remains elevated, indicating that this interaction may be influenced by additional regulatory factors.

Potential explanations include ineffective miRNA suppression due to post-transcriptional regulatory complexity, feedback loop dysfunction, or the presence of competing endogenous RNAs that sequester miR-34a. Additionally, developmental changes in the neonatal heart may modulate miR-34a and *HTR2A* interactions. Further studies are needed to clarify whether the miR-34a–*HTR2A* interaction reflects an adaptive or maladaptive response in cardiac hypertrophy.

**SGPP1.** Sphingolipids, including sphingosine-1-phosphate (S1P) and ceramide, act as lipid mediators in cardiomyocytes, responding to extracellular signals and regulating key intracellular pathways such as RAS-MAPK and PI3K-AKT [63]. *SGPP1* encodes a phosphatase primarily located in the endoplasmic reticulum that metabolises S1P into ceramide [64]. Elevated ceramide levels are associated with apoptosis and cell cycle arrest, whereas increased S1P activity promotes cell proliferation and inhibits apoptosis [65]. Although the role of sphingolipids in cardiovascular conditions such as myocardial infarction, hypertension, and stroke is not yet fully understood, their specific contribution to cardiac hypertrophy remains unclear [65]. This study reports *SGPP1* upregulation in 2-day-old HHRs, which may drive apoptosis and reduce cardiomyocyte proliferation during foetal development. This altered expression may be contributary to the reduced cardiomyocyte population at birth and the subsequent compensatory hypertrophy observed [12].

**GANC.** GANC belongs to glycoside hydrolase family 31 and encodes neutral α-glucosidase C, an enzyme implicated in glycogen metabolism [66]. The gene remains understudied; functional annotations in *Rattus norvegicus* are largely computational predictions; alpha-1,4-glucosidase, broader hydrolase activity and roles in carbohydrate metabolism; (National Center for Biotechnology Information, U.S. National Library of Medicine). During cardiac hypertrophy, myocardial metabolism shifts from fatty-acid oxidation to glycolysis, resembling the foetal metabolic state [8]. The potential role of neutral α-glucosidase C in these altered states remains unexplored.

Bioinformatic predictions suggest that *GANC* may be targeted by miR-351 and miR-490. However, RT-qPCR after transfection of H9C2(2−1) cells yielded inconclusive changes in *GANC* expression. In neonatal HHR hearts, RT-qPCR validation indicated a modest reduction in *GANC*, whereas microarray estimated an increase, indicating directional discordance. Together with the lack of supportive human genetic associations, these observations do not support a firm conclusion about *GANC* dysregulation in neonatal HHRs and should be interpreted with caution until confirmed by protein-level assays and functional validation studies.

**ITGA7.** *ITGA7* encodes the α7 subunit of integrins, mechano-receptors in myocytes that form heterodimers with β1 subunits and activate pro-hypertrophic PI3K-AKT and MAPK signalling upon ligand binding [37,67]. Overexpression of *ITGA7* promotes cellular regeneration and hypertrophy while reducing cardiomyopathy [68]. Additionally, α7β1 integrin overexpression preserves mitochondrial membrane potential under hypoxia, suggesting a role in mitigating cellular stress [69]. Human–murine evidence implicates ITGA7 loss of function in adult-onset cardiomyopathy; ITGA7$^{-/-}$ mice recapitulate conduction delays and systolic dysfunction, consistent with impaired α7β1-integrin mechano-transduction [70].

This study reports significant *ITGA7* downregulation in 2-day-old HHR hearts, potentially impairing the cardiomyocyte response to mechanical stress and weakening pro-survival signalling. As *ITGA7* supports mitochondrial stability, its reduced expression may increase vulnerability to oxidative stress and apoptosis, contributing to cardiomyocyte loss. This aligns with findings that HHRs have fewer cardiomyocytes at birth, requiring compensatory hypertrophy [12]. Interestingly, murine hearts subjected to transverse aortic constriction show increased *ITGA7* expression [71]. The discrepancy between upregulation in hypertensive models and downregulation in neonatal HHRs may reflect developmental or model-specific differences in response to mechanical stress. These findings underscore the need for further research to clarify how early *ITGA7* downregulation affects mechano-signal transduction and contributes to hypertrophic remodelling.

## 4. Comparison between microarray and RT-PCR results

The fold changes in gene expression validated by RT-PCR were compared to those reported in the microarray analysis. A key difference between the two datasets lies in the RNA source: the microarray analysis utilised RNA isolated from an earlier generation of 2-day-old HHRs, whereas RT-qPCR was performed on RNA from a more recent generation. This generational disparity may account for some of the observed variations, potentially due to genetic drift, a recognised phenomenon in inbred animal models.

While the possibility of technical error was considered, the observed increase in CWI in this study aligns with the elevated expression of established genetic markers of cardiac hypertrophy, including *NPPA*, *NPPB*, and *MAPK1*; a key component of the pro-hypertrophic RAS-MAPK signalling pathway. These consistent findings strengthen the validity of the candidate genes identified.

However, to confirm the biological or technical origins of the reported deviations in gene expression, further validation is required. This includes analysing RNA extracted from heart tissue of the current generation of neonatal HHRs. Such follow-up studies will help ascertain whether the observed discrepancies arise from generational changes or technical limitations, ensuring robust conclusions regarding gene expression profiles in the HHR model.

## 5. Future directions

Further investigation into the roles of miR-34a, miR-351, and miR-490*, alongside validated genes such as *HTR2A*, *SGPP1*, *ITGA7*, and *GANC* is essential to elucidate their contributions to early-onset cardiac hypertrophy. Extending these findings to human cardiac tissue would provide valuable translational insights. Validation by RT–qPCR was limited to a subset of microRNAs and genes because of resource constraints, with additional microarray-identified candidates deferred for future work. In place of external RNA-seq reanalysis, full gene-level results are provided in the Supplementary Data to ensure transparency and enable re-use. Confirming their expression and functional roles in cardiac hypertrophy is necessary. Evidence for GANC dysregulation remains uncertain in neonatal HHRs. It should be considered for protein-level confirmation and functional validation, with parallel evaluation in human cardiomyopathy datasets.

Novel genes and pathways identified through bioinformatics analyses also warrant detailed investigation to determine their contributions to cardiac disease and remodelling. Inconclusive RT-qPCR results for certain microRNA-target gene interactions suggest the need for alternative methodologies. Employing different cell lines or delivering microRNAs via plasmid vectors could enhance experimental reproducibility.

The disparity in CWI results in 2-day-old HHRs compared to previous studies underscores the need to re-characterise the current generation of neonatal HHRs to confirm the presence of hypertrophy and improve methods for quantifying hypertrophic changes beyond CWI metrics and markers belonging to FGP. Additionally, primary cardiomyocytes isolated from 2-day-old HHRs would provide a more physiologically relevant model for investigating miRNA-target interactions and downstream effects on apoptosis, proliferation, metabolism, and function, overcoming limitations of immortalised cell lines like H9C2-1.

Lastly, further exploration of the miR-34a–*HTR2A* regulatory interaction across different developmental stages in HHRs is recommended. Validating this pathway could offer critical insights into the mechanisms underlying cardiac hypertrophy and inform the development of targeted therapies, advancing our understanding of human cardiac hypertrophy and opening new treatment avenues.

## Conclusion

This study identifies three novel microRNAs; miR-34a, miR-351, and miR-490*, with altered expression in neonatal HHRs, linking them to key pathways associated with early-onset cardiac hypertrophy. GSEA highlighted processes related to apoptosis, metabolism, and myocardial remodelling. miR-34a was shown to regulate *HTR2A* in neonatal HHRs, consistent with serotonin-linked pathways in hypertrophic remodelling. Upregulation of *SGPP1* suggests enhanced sphingolipid

metabolism, potentially driving apoptosis and limiting cardiomyocyte proliferation. In contrast, downregulation of *GANC* and *ITGA7* points to disruptions in glycogen metabolism and impaired mechano-signal transduction, contributing to cardiomyocyte loss and compensatory hypertrophy. These findings underscore the role of altered miRNA expression in neonatal HHRs and highlight their influence on critical regulatory pathways. Further research is essential to determine the long-term impact of these regulatory changes and their relevance to human cardiac hypertrophy, paving the way for targeted therapeutic interventions.

## Supporting information

**S1 File. S1 Tables.** MicroRNA and Genes Microarray Results (HHR/NHR, 2-days). **S2 Tables**. Target genes and pathway enrichment analysis. **S1 Fig.** Number of KEGG Pathways potentially regulated by miRNAs under investigation either individually or communally. **S2 Fig.** Number of GO:BP potentially regulated by microRNAs under investigation either individually or communally. **S3 Fig.** Comparative analysis of predicted target genes regulated by miR-34a, miR-351, and miR-490*. **S4 Fig.** Raw Ct values from RT-PCR of h9C2-1 transfected with mimics and inhibitors.
(ZIP)

## Acknowledgments

The authors would like to thank the staff of the Deakin University Animal Facility for their assistance with animal care and tissue collection.

## Author contributions

**Conceptualization:** Shahzad Sadiq, Fadi J. Charchar, Tamsyn Crowley.

**Data curation:** Shahzad Sadiq, Andrew Sanigorski.

**Formal analysis:** Shahzad Sadiq, Andrew Sanigorski.

**Funding acquisition:** Fadi J. Charchar.

**Investigation:** Shahzad Sadiq.

**Methodology:** Shahzad Sadiq, Tamsyn Crowley.

**Project administration:** Shahzad Sadiq, Fadi J. Charchar.

**Resources:** Fadi J. Charchar, Tamsyn Crowley.

**Software:** Shahzad Sadiq, Tamsyn Crowley.

**Supervision:** Fadi J. Charchar, Tamsyn Crowley, Hansraj R. Bookun, David N. McClure.

**Validation:** Shahzad Sadiq, Andrew Sanigorski, Hansraj R. Bookun, David N. McClure.

**Visualization:** Andrew Sanigorski.

**Writing – original draft:** Shahzad Sadiq.

**Writing – review & editing:** Shahzad Sadiq, Fadi J. Charchar, Andrew Sanigorski, Tamsyn Crowley, Hansraj R. Bookun, David N. McClure.

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
