## [Decision Letter · Decision Letter 0]

8 Jul 2025

MicroRNA mediated regulation in early-onset cardiac hypertrophy: insights from the hypertrophic heart rat model

PLOS ONE

Dear Dr. Sadiq,

Thank you for submitting your manuscript to PLOS ONE. After careful consideration, we feel that it has merit but does not fully meet PLOS ONE’s publication criteria as it currently stands. Therefore, we invite you to submit a revised version of the manuscript that addresses the points raised during the review process.

Your paper was reviewed by an expert in the field and myself. As the reviewer pointed out, I would suggest the authors to add more  data  regarding gene expression.  In addition to those, genetic evidence may be feasible. I am aware of the journals` policy but this models were already established. 

We look forward to receiving your revised manuscript.

Kind regards,

Tomohiko Ai, M.D., Ph.D.

Academic Editor

PLOS ONE

2. Please include a caption for figure 3 and 4.

Additional Editor Comments (if provided):

Reviewers' comments:

Reviewer's Responses to Questions

**Comments to the Author**

1. Is the manuscript technically sound, and do the data support the conclusions?

Reviewer #1: Yes

2. Has the statistical analysis been performed appropriately and rigorously?

Reviewer #1: I Don't Know

3. Have the authors made all data underlying the findings in their manuscript fully available?

Reviewer #1: Yes

4. Is the manuscript presented in an intelligible fashion and written in standard English?

Reviewer #1: Yes

Reviewer #1: The authors presented the investigation of gene expression, using microarray technology, in the Hypertrophic Heart Rat (HHR) model compared to Normal Heart Rats (NHRs) and identified three miRNAs: miR-34a, miR-351, and miR-490* linked to key hypertrophic pathways including RAS-MAPK, PI3K-AKT, and calcium signaling. Furthermore, they validated their findings using RT-PCR and they transfected cell lines with mimic of those miR-34a, miR-351, and miR-490* to recapitulate their data.

The manuscript is well written and clear and the experiments are well conducted. However, the novelty of their findings is questionable.

The role of miR-34a has been previously established in cardiac remodeling, hypertrophy and fibrosis (PMID: 35155600, 36231079). In addition, Bcl2, Cyclin D1, Sirt1, PNUTS, and Notch1 are known targets of miR-34a (PMID: 26082557).

Similarly, role of both miR-351 (PMID: 36929003) and miR-490 (PMID: 32970185) in cardiac hypertrophy have been previously established as well.

The role of HRT2A has been previously established in promoting the development of cardiac hypertrophy by activating PI3K-PDK1-AKT-mTOR signaling (PMID: 32519137), and common variants in HTR2A may be associated with hypertension and metabolic syndrome in humans (PMID: 31906879).

The novelty of the work resides in the utilization of the HHR model in which they demonstrated upregulation of HRT2A and SGPP1, while it shows downregulation of GANC and ITGA7 (with the latter previously known as well, see: PMID: 36444867).

Therefore, the authors presented a nice replication in the HHR model of previously well established work in other mouse or rat models; this strengthen the data about gene expression pattern in cardiac hypertrophy.

The new result presented here is the GANC downregulation. However, little is known about the role of GANC expression and normal function in the heart and the authors did not present additional elucidative findings. In addition, there are no links between variants impairing the GANC function [i.e.: loss of function (LOF) variants] in humans with cardiac hypertrophy or heart failure. In gnomAD v.4.1 among the major constraint parameters, loss of function variants in GANC appear to be tolerated in the general population, suggesting that haploisufficiency may not be enough to trigger cardiac hypertrophy and a more severe concerted disruption maybe needed to exert a pathological effect. Despite the microarray data, RT-PCR validation could not confirm the downregulation, thus challenging the specificity of such result.

Further studies elucidating the role of GANC would be needed to strengthen the manuscript

**Do you want your identity to be public for this peer review?** For information about this choice, including consent withdrawal, please see our Privacy Policy

Reviewer #1: No

---

## [Author Response · Author response to Decision Letter 1]

29 Oct 2025

Rebuttal Letter

Re: PONE-D-25-28567

Title: MicroRNA-mediated regulation in early-onset cardiac hypertrophy: insights from the hypertrophic heart rat model

Corresponding author: Shahzad Sadiq

Thank you for the opportunity to revise our manuscript. We appreciate the constructive comments from you and the reviewer. In response, we have refined the manuscript’s framing, integrated the suggested literature with precise citations, standardised statistical reporting, and clarified gene-level interpretations—particularly for GANC. A Changes Table summarising section-by-section edits is included with this submission.

1) Framing and contribution

We recast the contribution as model-specific, defining miRNA–mRNA signalling in neonatal, normotensive HHRs, and removed/tempered “first/novel” phrasing in the Abstract, Introduction and Discussion.

2) Literature integration (per reviewer guidance; PMIDs shown)

1. miR-34a. We expanded the background and Discussion to integrate three key sources and to calibrate claims about therapeutic modulation and mechanism:

• Review (PMID 35155600): Summarises miR-34a’s roles in maladaptive remodelling/fibrosis and apoptosis; details repression of SIRT1, ALDH2, BCL2, PNUTS and links to genome-stability/telomere maintenance pathways. We used this to clarify mechanism and scope.

• Post-MI study (PMID 26082557): Shows miR-34a upregulation in injured myocardium and that anti-miR-34a improves LV function and reduces fibrosis and cardiomyocyte death after myocardial infarction, supporting a causal role in adverse remodelling.

• Diabetic cardiomyopathy study (PMID 36231079): Demonstrates that in established type 1 diabetic cardiomyopathy, anti-miR-34a confers only modest benefit, underscoring that efficacy varies by disease setting and stage.

• Manuscript changes: text revised in the miR-34a section of the Discussion to add these citations, articulate the model-/stage-dependence of therapeutic effects, and retain a conservative, context-specific framing of our neonatal HHR findings (no novelty claims).

2. miR-351. We incorporated evidence that miR-351, in concert with the long non-coding RNA LncSync, regulates cardiomyocyte homeostasis and hypertrophic growth in mice (PMID 36929003). This study provides developmental and hypertrophy-relevant context for miR-351 regulation, reinforcing its plausibility in our neonatal model.

• Manuscript changes: Expanded the miR-351 paragraph in the Discussion to cite 36929003 and to state that miR-351 participates in hypertrophic responses in experimental systems, with LncSync coordination in murine hearts.

3. miR-490*. We retained our cardiac statement that miR-490 constrains phenylephrine/RAS-driven hypertrophy via c-FOS/AP-1 (existing refs) and added non-cardiac mechanistic data showing that miR-490* represses telomere-maintenance genes (TERF2, TNKS2, SMG1), inducing telomere dysfunction and DNA-damage signalling (PMID 32970185). This broadens biological plausibility for miR-490 family involvement in growth-control programs.

• Manuscript changes: Added 32970185 and a one-sentence summary of the telomere-programme mechanism to the miR-490* section in discussion.

4. HTR2A (5-HT2A). We integrated mechanistic and human-genetic context: receptor activation engages PI3K–PDK1–AKT–mTOR signalling in an isoproterenol-induced hypertrophy model (PMID 32519137), and a large community-based study reported associations between HTR2A variants (including rs6313/T102C) and hypertension risk, with inter-cohort heterogeneity in effect direction (PMID 31906879).

• Manuscript changes: In the HTR2A subsection of the Discussion, added 32519137 to support the pathway statement and 31906879 to anchor human relevance, noting the cohort-to-cohort variability; retained our conservative interpretation of HTR2A upregulation and miR-34a regulation in our system (no priority claims).

5. ITGA7. We added translational evidence that ITGA7 loss of function is associated with adult-onset cardiomyopathy in humans and that Itga7⁻/⁻ mice recapitulate conduction delays and systolic dysfunction, consistent with impaired α7β1-integrin mechanotransduction (PMID 36444867).

• Manuscript changes: In the ITGA7 subsection, cited 36444867 and added a concise sentence linking our observed downregulation to mechanotransduction pathways and the human–murine phenotype, underscoring clinical relevance.

3) Results and statistics (precision & transparency)

• Exact p-values. Standardised to four decimals for all p ≥ 0.001 and defined star thresholds; updated throughout text and tables. (Example: Table 4 now shows GANC p = 0.0492.)

• GANC cross-platform note. Results now include a brief, non-interpretive statement flagging directional discordance (microarray ↑ vs RT-qPCR ↓) with a pointer to the Discussion comparison (classification as nominal/cautious).

4) Gene-level interpretation (selected examples)

• HTR2A. Increased in neonatal HHR hearts; cell experiments show miR-34a regulation of HTR2A. We integrate ISO-induced and pressure-overload contexts (mechanistic cascade and TAB model) and human cohort data, avoiding priority claims.

• SGPP1. Framed as consistent with increased sphingolipid metabolism/apoptosis in early remodelling.

• ITGA7. Downregulation aligned with perturbed mechano-transduction; added translational link via human–mouse genetics (PMID 36444867).

• GANC. Small-effect, borderline decrease by RT-qPCR (p = 0.0492) opposite to array estimate; limited human-genetic support and cross-platform mismatch → interpreted cautiously, with protein-level and functional validation deferred to future work.

5) Figures, tables, and nomenclature

• Figure 3 caption. Now reports star thresholds. Please note: Caption for Fig 3 was in the first manuscript (line: 208)

• Figure 4 caption. Please note: Caption for Fig 3 was in the first manuscript (line: 270)

• Global style fixes. Corrected RT-PCR → RT-qPCR, H9C2 → H9c2(2-1), unified star notation, corrected a miRNA table entry (rno-miR-490* p=0.0327), previously 0.327), and standardised gene/protein formatting and abbreviations at first use.

6) Data transparency and scope

Feasibility. We did not add new wet-lab experiments because we currently lack lab access, animals/samples, and funding; additional assays would require new approvals, reagents, and time beyond the revision window. Instead, to address the request for more gene-expression/genetic evidence while avoiding bias, we strengthened transparency and reporting as follows:

• Complete Supporting Information provided. We now include full differential-expression outputs as S1A Data — miRNA differential expression (HHR vs NHR) and S1B Data — mRNA differential expression (HHR vs NHR). Each table identifies the platform and normalization method, reports raw p-values and Benjamini–Hochberg FDR, and gives fold change (HHR/NHR; >1 = up in HHR). S1A also includes a nomenclature note mapping legacy “miR-490*” → miR-490-3p.

• Statistical transparency. We standardised exact p-values (for p ≥ 0.001) across text and tables, defined star thresholds once, and kept fold-change precision conservative. The GANC result is reported with exact statistics (p = 0.0492) and cross-referenced for array vs RT–qPCR directionality.

• Literature context and limitations. We integrated the key mechanistic and human-genetic papers highlighted by the reviewer, clarified limitations (including the small-effect/discordant GANC signal), and outlined feasible next steps (protein-level assays, functional perturbation, evaluation in human cardiomyopathy datasets).

• Bias control. We did not perform ad-hoc re-analysis of external RNA-seq datasets to avoid selective or inconsistently processed comparisons; instead, we prioritised complete reporting of our primary data in S1A/S1B.

We believe these revisions address the editorial and reviewer concerns and materially improve clarity, rigour, and transparency, consistent with PLOS ONE’s criteria. We appreciate your consideration.

Sincerely,

Shahzad Sadiq (on behalf of all authors)

---

## [Decision Letter · Decision Letter 1]

30 Nov 2025

MicroRNA mediated regulation in early-onset cardiac hypertrophy: insights from the hypertrophic heart rat model

PONE-D-25-28567R1

Dear Dr. Sadiq,

We’re pleased to inform you that your manuscript has been judged scientifically suitable for publication and will be formally accepted for publication once it meets all outstanding technical requirements.

Kind regards,

Tomohiko Ai, M.D., Ph.D.

Academic Editor

PLOS ONE

Additional Editor Comments (optional):

Reviewers' comments:

Reviewer's Responses to Questions

**Comments to the Author**

Reviewer #1: All comments have been addressed

2. Is the manuscript technically sound, and do the data support the conclusions?

Reviewer #1: Yes

3. Has the statistical analysis been performed appropriately and rigorously?

Reviewer #1: I Don't Know

4. Have the authors made all data underlying the findings in their manuscript fully available?

Reviewer #1: Yes

5. Is the manuscript presented in an intelligible fashion and written in standard English?

Reviewer #1: Yes

Reviewer #1: The authors have adequately addressed the reviewers' comments and the manuscript has been improved compared to the previous submission

**Do you want your identity to be public for this peer review?** For information about this choice, including consent withdrawal, please see our Privacy Policy

Reviewer #1: No

---

## [Editor Report · Acceptance letter]

PONE-D-25-28567R1

PLOS One

Dear Dr. Sadiq,

I'm pleased to inform you that your manuscript has been deemed suitable for publication in PLOS One. Congratulations! Your manuscript is now being handed over to our production team.

Kind regards,

on behalf of

Dr. Tomohiko Ai

Academic Editor

PLOS One